# An Optical Bridge is Enough: Cross-Resolution Microdisk Code Classification without Super-Resolution

**Changwoo Kim**                                          CWKIM@KETI.RE.KR

**Seong-Eun Kim**[*]                                       SEKIM@KETI.RE.KR

*Korea Electronics Technology Institute (KETI), Seongnam-si, Republic of Korea*

## Abstract

Hole-encoded microdisks ($\sim 50\,\mu$m) enable multiplexed in-vitro diagnostics, but their code patterns are barely visible under widely deployed 4× microscopes. Optical Bridge Domain Adaptation (OBDA) trains on high-resolution equipment crops, a small 10× bridge set and stochastic degradation, reaching 98.26% accuracy on unseen 4× crops without any 4× labels. Degradation synthesis, test-time augmentation and bridge data contribute +57.7, +0.95 and +1.27 pp, respectively. OBDA is 2.4 pp more accurate than the strongest of six super-resolution baselines and 27× faster at inference. These preliminary four-code results motivate an in-progress extension to the full 71-code catalogue.

**Keywords:** domain adaptation, cross-resolution classification, microscopy, multiplexed diagnostics, stochastic degradation.

## 1. Introduction

Coded-particle multiplexed in-vitro diagnostics (IVD) identify physical tags on particles to read several biomarkers per sample (Kelley et al., 2014; Pregibon et al., 2007). We study silica-coated microdisks ($\sim 50\,\mu$m) encoded by etched hole patterns, using four codes (#56, #68, #75, #79), under three optical regimes with substantial domain shift (Stacke et al., 2021) (Figure 1): dedicated analysis equipment (Equip.), where holes are unambiguous; 10× microscopes, where patterns remain readable; and widely deployed 4× objectives, where the signature nearly disappears. Because dedicated equipment is expensive and unavailable at many satellite sites, reliable 4× classification is the deployment bottleneck.

A natural baseline is to restore 4× inputs with super-resolution (SR) before classification. Here, however, all six SR pipelines we evaluated—RRDB (Wang et al., 2018), CUT (Park et al., 2020), ZSSR (Shocher et al., 2018), DDPM (Ho et al., 2020), classifier-guided restoration and frequency-separated restoration—reduced accuracy relative to direct low-resolution classification, suggesting that visually plausible reconstruction is less useful than learning resolution-robust discriminative cues (Dai et al., 2016). Feature-alignment adaptation (Ganin et al., 2016; Long et al., 2015; Kamnitsas et al., 2017) typically relies on abundant unlabelled 4× imagery at training time; we instead use only 4× amplitude statistics via Fourier domain adaptation (FDA) (Yang and Soatto, 2020).

We use the 10× domain as an optical bridge: it shares the microscope family with 4× while retaining readable hole patterns, forming an intermediate resolution regime between the endpoints, analogous to curriculum learning (Bengio et al., 2009). Mixing modest 10× data with aggressive degradation of equipment crops reaches 98.26% on unseen 4×,

---

[*] Corresponding author.

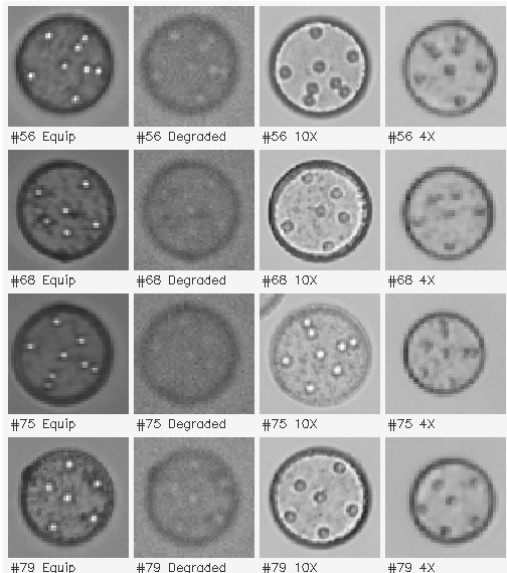

Figure 1: Microdisk crops across regimes: dedicated analysis equipment (Equip.), degraded equipment, $10\times$ microscope and $4\times$ microscope for codes #56, #68, #75 and #79. Holes are clear on Equip., readable at $10\times$ and nearly invisible at $4\times$.

without $4\times$ labels. Contributions: (i) **OBDA**, using only 39 bridging $10\times$ acquisitions; (ii) a **systematic negative result** for six SR methods; and (iii) a **factorial decomposition** of degradation, bridge data and test-time augmentation (TTA).

## 2. Method

**Data and classifier.** Training combines 36,000 equipment crops (sub-sampled from 2,304 images) and 1,078 $10\times$ crops from 39 acquisitions (oversampled to 9,220/epoch); 40 held-out $4\times$ acquisitions (5,979 crops) form the test set, disjoint at the source-image level. Disks are localised by a multi-threshold fusion detector ($+85/237/115\%$ detections on Equip./$10\times$/$4\times$ over single thresholds). Cls96 is a five-stage CNN with squeeze-and-excitation blocks (Hu et al., 2018) ($\sim$14.6M parameters) on $96\times96$ crops, trained with AdamW, focal loss, CutMix and cosine-with-restart for 60 epochs; inference uses 8-way TTA.

**Optical Bridge DA.** OBDA minimises empirical risk on $\mathcal{D}_E \cup \mathcal{D}_{10}$ under stochastic degradation $g$: Gaussian blur, downscale–upscale cycling, FDA (Yang and Soatto, 2020) to $4\times$ amplitude and Poisson–Gaussian noise, applied to equipment crops during training with probability $p_{\mathrm{deg}}{=}0.7$. The $10\times$ bridge matches $4\times$ in illumination ($|\Delta\mu|{\leq}3.6$) and Equip. in feature geometry, with centroid cosines of 0.995 ($10\times$–Equip.), 0.974 ($10\times$–$4\times$) and 0.968 (Equip.–$4\times$). This combination forces resolution-robust texture rather than absolute brightness ($\sim$60-unit $\mathcal{D}_E$–$\mathcal{D}_4$ gap), while degradation closes the residual gap so crops cluster by

Table 1: Factorial decomposition on the $4\times$ test set (5,979 crops). Values are classification accuracy (%). *Deg./Br./TTA*: degradation synthesis, $10\times$ bridge, 8-way TTA; AdaBN adapts BN statistics at test time.

| Configuration | Deg. | Br. | TTA | $4\times$ (%) | $10\times$ (%) |
|---|---|---|---|---|---|
| Equipment only | – | – | ✓ | 38.30 | 63.54 |
| AdaBN (Li et al., 2018) | – | – | ✓ | 78.51 | – |
| + Degradation | ✓ | – | – | 96.04 | 98.79 |
| + TTA | ✓ | – | ✓ | 96.99 | 98.98 |
| **+ Bridge (OBDA)** | ✓ | ✓ | ✓ | **98.26** | **99.81** |

code rather than domain. BN rescaling alone is insufficient (Table 1, AdaBN row; (Li et al., 2018)).

## 3. Results

On a controlled 2,777-crop $4\times$ subset, direct OBDA reaches 98.19%, whereas all six SR baselines underperform: ZSSR 95.75% at $27\times$ slower, CUT 94.06%, RRDB 80.91%, classifier-guided 71.84%, frequency-separated 71.70% and DDPM 28.38%; a domain-specific RRDB fine-tune did not close the gap. These results suggest that SR pixel fidelity can smooth the sub-pixel texture cues used at $4\times$ (disk radius $\sim$17 px). Table 1 decomposes the full 5,979-crop test set: equipment-only collapses at 38.30%, AdaBN (Li et al., 2018) reaches 78.51%, degradation synthesis dominates (+57.7 pp), TTA adds +0.95 pp and the $10\times$ bridge adds +1.27 pp (42% error reduction) without harming $10\times$. The 95% bootstrap CI for 98.26% is $[97.90, 98.59]$%, and a confidence gate gives 99.0% accuracy at 94.8% coverage. The classifier is overconfident (ECE 0.29) and will require post-hoc temperature calibration. An INT8-quantised quarter-scale distillation of CLS96 (0.91 M params, 98.13% on $4\times$) projects to $\sim$15 FPS on a fanless Jetson-class edge SoC (host-scaled estimate, on-device validation pending), suggesting that $4\times$ deployment can also keep compute cost low.

## 4. Conclusion

OBDA combines equipment crops, a $10\times$ bridge and heavy degradation to classify $4\times$ microdisk codes without $4\times$ labels, outperforming six SR pipelines (98.26%, $27\times$ faster). For microdisk coding, data-centric bridging gives stronger cross-resolution robustness than image restoration. Extension to the full 71-code catalogue is the next step.

## Acknowledgments

This work was supported by the Korea Planning & Evaluation Institute of Industrial Technology (KEIT) grant funded by the Ministry of Trade Industry and Resources (MOTIR) (No. RS-2025-14383307, Development of a Multimodal On-Site Diagnostic System for Rapid Identification of Infectious Bacteria and Antibiotic Susceptibility Testing).

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
