# OpenReview forum: "An Optical Bridge is Enough: Cross-Resolution Microdisk Code Classification without Super-Resolution"
_MIDL.io/2026/Short_Papers — MIDL 2026 - Short Papers Poster_

### Official Review · Reviewer_Bkvp · 2026-05-03
**Method for Microdisk Code Classification on 4x microscopes images without using super-resolution before the classification.**

**Rating:** 3
**Confidence:** 3

**Review:**

This paper presents a method for Microdisk Code Classification on 4x microscopes images without using super-resolution before the classification. Instead of using super-resolution, they use an Optical Bridge Domain Adaptation (OBDA), which trains on high-resolution equipment crops, a small 10x bridge set and stochastic degradation. The OBDA method
The stochastic degradation g  is composed of Gaussian blur, downscale-upscale cycling, Fourier-domain adaptation, to 4x amplitude, Poisson-Gaussian noise.

**Summary:**

This paper presents a method for Microdisk Code Classification on 4x microscopes images without using super-resolution before the classification. Instead, they use an Optical Bridge Domain Adaptation (OBDA), which trains on high-resolution equipment crops, a small 10x bridge set and stochastic degradation.
Experiments show better results than classical super-resolution methods. A factorial decomposition attributes the main gain to degradation synthesis.

**Strengths:**

This paper presents a new method for Microdisk Code Classification on 4x microscopes images without using super-resolution before the classification. This methods obtains better results than classical super-resolution methods.

**Weaknesses:**

Some parts of the paper are not clear. Some abbreviations are not defined (for example TTA, Test-Time augmentation), which makes reading very difficult.
The OBDA framework is not detailed.
Which metric is presented in table 1?
It might have been interesting to test different values of the probability pdeg on the results.

**Justification Of Rating:**

Proposed method seems interesting ans shows good results but some information and explanation on the proposed OBDA method are missing.

---

### Decision · Program_Chairs · 2026-05-08

Accept (Poster)